# Unsupervised Insurance Fraud Prediction Based on Anomaly Detector Ensembles

## Alexander Vosseler

Allianz Global Corporate & Specialty SE (AGCS), 85774 Unterföhring, Germany; alexander.vosseler@allianz.com

**Abstract:** The detection of anomalous data patterns is one of the most prominent machine learning use cases in industrial applications. Unfortunately very often there are no ground truth labels available and therefore it is good practice to combine different unsupervised base learners with the hope to improve the overall predictive quality. Here one of the challenges is to combine base learners that are accurate and divers at the same time, where another challenge is to enable model explainability. In this paper we present BHAD, a fast unsupervised Bayesian histogram anomaly detector, which scales linearly with the sample size and the number of attributes and is shown to have very competitive accuracy compared to other analyzed anomaly detectors. For the problem of model explainability in unsupervised outlier ensembles we introduce a generic model explanation approach using a supervised surrogate model. For the problem of ensemble construction we propose a greedy model selection approach using the mutual information of two score distributions as a similarity measure. Finally we give a detailed description of a real fraud detection application from the corporate insurance domain using an outlier ensemble, we share various feature engineering ideas as well as discuss practical challenges.

**Keywords:** Bayesian anomaly detection; outlier ensembles; insurance claims fraud; unsupervised learning; model explanation

## 1. Introduction

The detection of outliers or anomalous data patterns is one of the most prominent machine learning use cases in the industry. Applications range from quality control, intrusion detection, web log analytics to medical applications. In the finance industry the prediction of credit card fraud (cf. Buonaguidi et al. 2022), stock market anomalies and insurance claim fraud (cf. Gomes et al. 2021) are the most common examples of anomaly detection (see Phua et al. 2010 for a survey of fraud detection). Statistically an anomalous observation (or an outlier) could be defined as "[...] an observation which deviates so much from other observations as to arouse suspicions that it was generated by a different mechanism." (Hawkins 1980). Some popular methods for outlier detection are based on the distance between observations (cf. Angiulli and Pizzuti 2002; Knorr and Ng 1997), others are based on the variance of angles between sample points in high dimensional feature spaces (Kriegel et al. 2008) or use the number points in specific regions of the space ("density-based") to define outliers (cf. Aggarwal 2012; Breunig et al. 2000; Papadimitriou et al. 2003). Since methods for outlier detection are mostly unsupervised this makes model diagnosis (or evaluation) and model selection challenging due to the lack of ground truth. For this reason instead of selecting a single "best" model it is common practice to use ensemble approaches instead (cf. Zimek et al. 2014). Ensemble learning uses various base detectors to achieve a better predictive accuracy compared to using a single base detector. Those methods involve two main steps: (i) creating a candidate set of base detectors and (ii) combining the base detectors to get an improved model score. In general the balance of accuracy and diversity of the involved base detectors is crucial to form good model ensembles (cf. Schubert et al. 2012), since we would like to combine

models that are accurate, but make different errors on a given data set ("diversity"). Since ensemble learning often means combining many black-box models with each other the task of model explanation (or interpretability) becomes even more challenging. In many industrial applications, like in banking or insurance, model interpretability is crucial due to regulatory requirements.

In this paper we will focus on the application of outlier ensembles for the detection of insurance claim fraud. For this purpose we use a combination of different approaches, like generative neural networks, density-based approaches, kernel methods and tree-based methods among others. In this context we will first introduce an own probabilistic anomaly detection method in Section 2, which will be one component in our final outlier ensemble used to detect insurance fraud. For this the work presented in Knuth (2019) on Bayesian density estimation will be extended to the domain of anomaly detection. For this purpose and in contrast to the aforementioned work an informative hierarchical prior for the unknown number of bins is introduced and the posterior predictive distribution is used to formulate a Bayesian anomaly detection algorithm (BHAD). BHAD scales linearly with the size of the data and allows a direct explanation of individual anomaly scores due to its simple linear functional form, which makes it very suitable for practical applications when model interpretability is crucial. Other histogram-based outlier approaches like in Koufakou et al. (2007) or Goldstein and Dengel (2012) are on the other hand not based on a probabilistic framework and hence do not allow direct statistical inference regarding the unknown quantities, like the number of histogram bins, which is estimated as a step of BHAD (see Section 2). Although not directly related to the aforementioned model there is a growing literature of unsupervised probabilistic methods for fraud detection, see Ekin et al. (2019); Zafari and Ekin (2019).

In a simulation study (Section 4) and also using two popular benchmark datasets (Section 5) we analyze the predictive performance of the used candidate models and also compare them with different model ensembles. The results suggest that the proposed BHAD model has very competitive performance compared to other more complex models like variational autoencoders, in fact it is among the best performing candidates while offering individual score interpretability. Since selecting accurate and divers candidates is crucial to form powerful model ensembles we use a variation of the original Greedy model selection approach (Schubert et al. 2012) in Section 3.2 using the more general mutual information of two score distributions as a selection criterion instead of a (weighted) Pearson correlation as originally suggested by the authors.

Since providing model explanations can be a challenging task, especially in an ensemble of unsupervised black box models we propose a model-agnostic approach for such situations in Section 3.3. The idea is to fit a meta (or surrogate) model to a "pseudo target", i.e., the predicted output of the model ensemble, and then regress it onto the original feature variables. The approach allows using standard regression and classification techniques as a global model explainer, which can be combined with any state-of-the-art model explanation approach like LIME, SHAP etc. To the best of our knowledge, this approach although simple and straightforward to implement has not been proposed in the outlier detection literature yet.

In Section 6 an outlier ensemble model using Greedy model selection of base detectors is then utilized to detect fraudulent insurance claims using data from the corporate insurance domain. Finally in Section 6.2 we present some empirical results of a corporate insurance industry application for five countries. For the latter task also various ideas for the extraction of predictive features are shared with the readers. For example, building features based on an own fuzzy names matching algorithm is outlined as well as using different NLP approaches for handling unstructured claim descriptions. In this context we also discuss common data quality issues in practice regarding valid labels which could give some guidance to practitioners working in the field of insurance fraud detection. Section 7 summarizes the results and concludes.

To summarize, in this paper we make the following contributions:

- we present BHAD, a novel Bayesian histogram-based anomaly detection method, which directly estimates the unknown number of bins using an informative hierarchical prior
- we introduce a generic approach to enable local and global model explanations for outlier ensembles using a supervised surrogate model
- we propose a variation of the Greedy model selection algorithm of Schubert et al. (2012) using the mutual information of two score distributions as a similarity measure
- we give a detailed description of a real insurance claims fraud detection application and share various feature engineering ideas, e.g., utilizing natural language processing and clustering techniques

Next we will outline a Bayesian anomaly detector based on a hierarchical Categorical-Dirichlet mixture approach which can be used for continuous and/or categorical features.

## 2. Bhad: Bayesian Histogram-Based Anomaly Detector

### 2.1. Likelihood Function

First let's assume we observe an i.i.d. sample of scalar real-valued observations $y_i, i = 1, \ldots, N$ and we want to estimate the unknown probability density function (p.d.f.). We will first present a Bayesian version of a univariate histogram estimator and then show how this can be used in the context of multivariate anomaly detection.

Assume the following piecewise-constant data model (cf. Knuth 2019) for the $i$-th observation $y_i \in \mathbb{R}$:

$$f(y_i|\pi, \xi, K) = \sum_{k=1}^{K} \mathbf{1}(\xi_{k-1} \le y_i < \xi_k) \cdot \frac{\pi_k}{\nu_k} \tag{1}$$

with $y_i \in [\xi_0, \xi_K], \forall i$, where $\xi_0 < \xi_1 < \cdots < \xi_K$ denote the knots which we will assume as known for simplicity as well as the number of bins $K$. Also let $\nu_k = \xi_k - \xi_{k-1}$ denote the bin width, which will be assumed equal across bins, i.e., $\nu_k = \nu, \forall k$.[1]

Let $\pi_k = \Pr(y_i \in B_k)$, $k = 1, \ldots, K$, denote the (unknown) probability of the observation $i$ falling into the $k$-th bin, with $B_k = [\xi_{k-1}, \xi_k)$.[2]

For a model with $K$ bins this constitutes the entire range of the observed data $V = K \cdot \nu$ (Ibid.). For the vector $\mathbf{y} \equiv (y_1, \ldots, y_N)$ the joint data density function is given as:

$$f(\mathbf{y}|\pi, \xi, K) = \prod_{k=1}^{K} \prod_{i:y_i \in B_k}^{N} \left(\frac{K}{V}\right) \cdot \pi_k \tag{2}$$

For brevity let $\xi \equiv (\xi_0, \xi_1, \ldots, \xi_K)$ and $\pi \equiv (\pi_1, \ldots, \pi_K)$ be the vector of knots and bin probabilities, respectively.

Next define an auxiliary variable $z_i^{(k)} \equiv \mathbf{1}(y_i \in B_k) \in \{0, 1\}$ to indicate the assignment of $y_i$ to one of the $K$ categories, i.e., $y_i$ is mapped to a $K$-dimensional "one-hot" vector $z_i$ with one in the $k$-th component and otherwise zeros. From Equation (2) it can be recognized that the sample density function, as a function of $\pi$, has the form of a categorical (or multinoulli) $\mathrm{Cat}(\pi_1, \ldots, \pi_K)$ distribution with p.m.f. $f(z_i^{(k)} = 1) = \pi_k$ and hence the joint data density is a multinomial distribution with sufficient statistics $n_k = \sum_{i=1}^{N} z_i^{(k)}$, where $N = \sum_{k=1}^{K} n_k$. For the subsequent analysis we define an $N \times K$ sparse indicator matrix $\mathbf{Z}$ with row vectors $z_i$, $i = 1, \ldots, N$ and represent the data $\mathbf{y}$ in terms of the matrix $\mathbf{Z}$ instead.

### 2.2. Prior Distributions

In a Bayesian context the unknown model parameters are treated as random variables (rather than fixed and unknown population quantities) to which prior distributions are assigned in order to express the available knowledge about the statistical task. In the following we treat the bin probabilities $\pi_k, k = 1 \ldots K$, as well as the number of bins, $K$, as random model parameters on which we would like to do inference.

Conditional on $K$, the likelihood function of $\pi$ in Equation (2) has the form of a multinomial distribution and hence we could use a conjugate Dirichlet prior, $\text{Dir}(\pi|\alpha_1, \ldots, \alpha_K)$, with hyperparameters $\alpha_k > 0, \forall k$. To express lack of prior knowledge we use the corresponding objective Jeffreys prior (see Jeffreys 1961) with $\alpha_k = 1/2, \forall k$:

$$p(\pi|K) = \frac{\Gamma(\frac{K}{2})}{\Gamma(\frac{1}{2})^K} \cdot \prod_{k=1}^{K} \pi_k^{-1/2} \tag{3}$$

with $\Gamma(.)$ denoting the Gamma function.

Next we would like to express our prior knowledge regarding the number of bins $K$. Assuming a flat prior would assign the same prior probability to all values, which does not seem suitable for all contexts. Following the principle of parsimony we therefore prefer simpler over complicated models, i.e., a model with less number of bins over a model with many number of bins. Subsequently we use the geometric prior probability mass function of Scargle et al. (2013):

$$p(K|\gamma; K_{max}) = c_0 \cdot \gamma^K \, , \quad K = 1, \ldots, K_{max} \tag{4}$$

with prior hyperparameter $\gamma$ and normalization constant $c_0$. The latter can be derived using basic properties of the geometric series $\sum_{j=0}^{J} \gamma^j = \frac{1 - \gamma^{J+1}}{1 - \gamma}$ for $\gamma \neq 1$. Note this prior assigns more weight to models with fewer bins for $0 < \gamma < 1$ and the smaller $\gamma$ the more pronounced this weighting effect will be.

As the prior in (4) is sensitive to choices in $\gamma$, we also model this hyperparameter explicitly by assigning a uniform prior density $U_\gamma(0,1)$ to it, which leads to a hierarchical prior for the number of bins. This is in contrast to the aforementioned authors who treat this hyperparameter as known in their analysis.

This leads to the joint prior for the number of bins $K$ and the hyperparameter $\gamma$:

$$p(K, \gamma|K_{max}) = \frac{1 - \gamma}{1 - \gamma^{K_{max}+1}} \cdot \gamma^K \cdot U_\gamma(0,1) \tag{5}$$

### 2.3. Posterior Distributions

Combining the prior with the likelihood in Equation (2) using Bayes theorem and noting that the likelihood is independent of $\gamma$ the joint posterior is given by (omitting conditioning on $\xi$ and $K_{max}$ subsequently):

$$p(\pi, K, \gamma|\mathbf{Z}) \propto p(\pi|K) \cdot p(K, \gamma) \cdot f(\mathbf{Z}|\pi, K) \tag{6}$$

In the outlier analysis below we will condition on a given number of bins per feature as well as on a $\gamma$ value. For this we will compute the maximum-a-posteriori (MAP) estimates $(\hat{K}_{MAP}, \hat{\gamma}_{MAP}) = \arg\max_{K,\gamma} p(K, \gamma|\mathbf{Z})$ based on the joint posterior $p(K, \gamma|\mathbf{Z})$. We first compute $\hat{\gamma}_{MAP}$ based on the marginal posterior of $\gamma$:

$$p(\gamma|\mathbf{Z}) = \sum_{k=1}^{K_{max}} p(K = k, \gamma) \cdot f(\mathbf{Z}|K = k) \tag{7}$$

with $f(\mathbf{Z}|K)$ the marginal likelihood of $K$, i.e., after analytically integrating out the bin probabilities $\pi$ using properties of the Dirichlet distribution (cf. Knuth 2019; Murphy 2012):

$$f(\mathbf{Z}|K) = \int p(\pi|K) \cdot f(\mathbf{Z}|\pi, K) \cdot dS_K \tag{8}$$

$$= \frac{\Gamma(\sum_{k=1}^{K} \alpha_k)}{\prod_{k=1}^{K} \Gamma(\alpha_k)} \cdot \left(\frac{K}{V}\right)^N \cdot \int \prod_{k=1}^{K} \pi_k^{\sum_{i=1}^{N} z_i^{(k)} + \alpha_k - 1} \cdot dS_K \tag{9}$$

$$= \left(\frac{K}{V}\right)^N \cdot \frac{\Gamma(\frac{K}{2})}{\Gamma(\frac{1}{2})^K} \cdot \frac{\prod_{k=1}^K \Gamma(n_k + \frac{1}{2})}{\Gamma(N + \frac{K}{2})} \tag{10}$$

with $\mathcal{S}_K$ denoting the $(K-1)$-dimensional simplex with $(\pi_1, \ldots, \pi_K) \in \mathcal{S}_K$, i.e., the support of the Dirichlet distribution and $n_k = \sum_{i=1}^N z_i^{(k)}$. Note that the last equation follows from using the normalizing constant of the Dirichlet distribution and assuming a Jeffreys prior for $\pi$, i.e., $\alpha_k = 1/2, \forall k$.

Then the conditional posterior mass function of $K$ given $\hat{\gamma}_{MAP}$ is computed using Equations (4) and (8) from which $\hat{K}_{MAP}$ can then be calculated:

$$p(K|\hat{\gamma}_{MAP}, \mathbf{Z}) = p(K|\hat{\gamma}_{MAP}) \cdot f(\mathbf{Z}|K) \tag{11}$$

The posterior of the bin probabilities follows from the conjugacy of the multinomial distribution and the corresponding Jeffreys prior and is given by

$$p(\pi|K, \mathbf{Z}) \propto \prod_{k=1}^K \pi_k^{\alpha_k - 1} \cdot \prod_{i: z_i^{(k)}=1}^N \frac{\pi_k}{v_k} \propto \prod_{k=1}^K \pi_k^{n_k - 1/2} \tag{12}$$

with sufficient statistics $n_k = \sum_{i=1}^N z_i^{(k)}$ as above and hence has the well-known form of a $\text{Dir}(\pi|\alpha_1^\star, \ldots, \alpha_K^\star)$ p.d.f. with updated posterior parameters $\alpha_k^\star = n_k + 1/2$.[3]

For illustration purposes we have generated an independent $N = 7000$ sample from a multi-modal Cauchy mixture distribution, see Figure 1. The corresponding posteriors of the number of bins and of $\gamma$ are depicted in Figure 2 from which the MAP estimates $\hat{K}_{MAP} = 54$ and $\hat{\gamma}_{MAP} = 0.68$ were computed.[4]

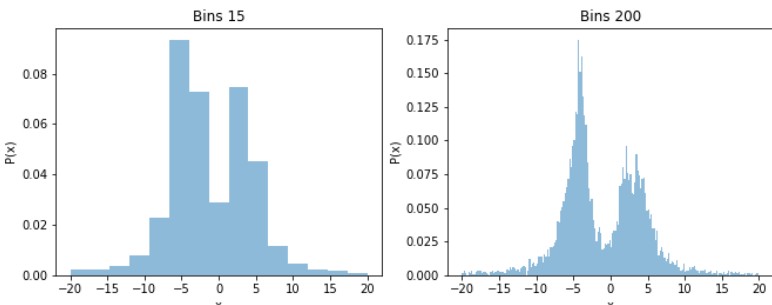

**Figure 1.** Cauchy data with density estimates.

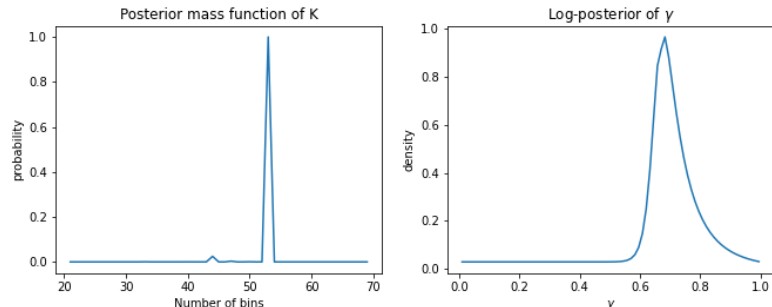

**Figure 2.** Posteriors of number of bins (**left**) and of $\gamma$ (**right**).

### 2.4. Anomaly Prediction

For a new data point $z_{N+1}$ we need the posterior predictive distribution $f(z_{N+1}|z_1, \ldots, z_N)$ in order to evaluate the probability of the event $z_{N+1}^{(s)} = 1$ with $s = 1, \ldots, K$. Conditional on MAP estimates $\hat{\gamma}$ and $\hat{K}$ and after analytically integrating out the bin probabilities $\pi$ the

posterior predictive probability of $z_i^{(s)} = 1$ has the following simple form (cf. Murphy 2012, p. 83):

$$f(z_{N+1}^{(s)} = 1|\hat{K}, \hat{\gamma}, \mathbf{Z})$$

$$= \int f(z_{N+1}^{(s)} = 1|\pi, \hat{K}, \hat{\gamma}) \cdot p(\pi|\hat{K}, \mathbf{Z}) \cdot \mathrm{d}\mathcal{S}_{\hat{K}} \tag{13}$$

$$\propto \frac{\alpha_s + n_s}{\sum_{k=1}^{\hat{K}} (\alpha_k + n_k)}$$

with $\alpha_k = 1/2, \forall k$ when using the Jeffreys prior, the sufficient statistics $n_k$ of the multinomial distribution and $\mathcal{S}_K$ the support of the Dirichlet distribution.

We can next use the above to construct a simple but efficient unsupervised anomaly detector based on the posterior predictive probabilities in Equation (13). Let $\mathbf{x}_i \equiv [x_{i,1}, \dots, x_{i,D}]^T$, $i = 1, \dots, N$, be a feature vector where the $D$ independent features can have mixed scale level (e.g., categorical, ordered, continuous etc.).[5] In the case of continuous features we partition their support into $K_j, j = 1 \dots D$, disjoint intervals as described above, whereas in case of discrete features each level constitutes its own histogram bin.

As anomaly score for observation $i$ we will use the average posterior predictive probabilities over the feature space in Equation (13). It can be shown using Bayesian reasoning (see Nelson 2017) that for a given set of probabilities the geometric mean represents the correct average probability and hence we will use the geometric average of the posterior predictive probabilities (assuming feature independence) as anomaly score or equivalently the arithmetic average in log-space:

$$S(\mathbf{z}_i) = \frac{1}{M} \sum_{j=1}^{D} \sum_{s_j=1}^{K_j} \log(f(z_{i,j}^{(s_j)} = 1|\hat{K}_j, \hat{\gamma}_j, \mathbf{Z}_j)) \tag{14}$$

with $z_{i,j}^{(s_j)} = 1$ if for feature $j$ the original observation $i$ is assigned to the $s_j$-th interval with $s_j = 1 \dots K_j$, $\mathbf{z}_i$ is a $M$-dimensional binary vector for the bin assignments, with $M = \sum_{j=1}^{D} K_j$, and $\mathbf{Z}_j$ denotes an $N \times K_j$ sparse matrix with $K_j$ dimensional binary row vectors $z_{i,j}$.

We call this model: Bayesian Histogram Anomaly Detector (BHAD). The used Bayesian density estimator is in contrast to the one used in Knuth (2019) who does not use the feature independence restriction when computing the multi-dimensional histograms, but at higher computational costs which often can be a problem in industrial applications. Also in contrast to the aforementioned work since our goal is anomaly detection our main quantity of interest is the predictive distribution, which is not of primary relevance when the goal is pure density estimation and hence inference. The steps to compute BHAD are sketched in Algorithm 1.

It is interesting to observe that the proposed Bayesian anomaly detector bears some resemblance to other works like for example the Attribute Value Frequency (AVF) algorithm of Koufakou et al. (2007), if we set the prior hyperparameters $\alpha_k = 0, \forall k$. However the AVF algorithm was proposed for the use of categorical data only, whereas BHAD can be applied to categorical and numerical features.[6] Another detection method similar to the one presented here is the "Histogram-based outlier score (HBOS)" of Goldstein and Dengel (2012). However since BHAD directly computes the posterior of the unknown parameters $(\pi_1, \dots, \pi_K)$, $K$ and $\gamma$ we not only get point estimates for these quantities, but a whole distribution as a measure of uncertainty. This is in sharp contrast to the two aforementioned works, which are not based on an explicit probabilistic framework and hence do not allow straightforward statistical inference, like for example the computation of predictive bounds for the number of bins $K$. Also note that setting $\alpha_k \neq 0$ in BHAD this corresponds to Laplace smoothing through the introduction of the "pseudo counts" $\alpha_k$ and hence to smoother or less sparse score distributions compared for example to the AVF algorithm.

---

**Algorithm 1:** Bayesian Histogram-based Anomaly Detector

---

/* Set maximum number of bins $K_{Max}$ for all features */;
$j \leftarrow 1$;　　　　　　　　　/* Features */
$\mathcal{T} := \varnothing$;　　　　　　　　/* Sufficient statistics */
**while** $j \leq D$ **do**
　　Compute $p(\gamma_j | \mathbf{Z}_j)$ using Equations (7) and (8);
　　Compute $\hat{\gamma}_j = \arg\max_\gamma p(\gamma_j | \mathbf{Z}_j)$;
　　Compute $p(K_j | \hat{\gamma}_j, \mathbf{Z}_j)$ using Equation (11);
　　Compute $\hat{K}_j = \arg\max_K p(K_j | \hat{\gamma}_j, \mathbf{Z}_j)$;
　　Compute sufficient statistics $\tau_j = \{n_1, \ldots, n_{\hat{K}_j}\}$ based on $\mathbf{Z}_j$;
　　Compute posterior predictive prob. in Equation (13) based on $\mathbf{Z}_j$ using $\tau_j$ ;
　　$\mathcal{T} \leftarrow \mathcal{T} \cup \tau_j$ ;
　　$j \leftarrow j + 1$;
**end**
Compute *Scores* $= (S(\mathbf{z}_1), \ldots, S(\mathbf{z}_N))$ using Equation (14);
**return** *Scores*;

---

After having assigned a score to each sample in the data set, using expression (14), we calculate a left-tail decision threshold as the $p\%$—quantile (e.g., 1%) of the score distribution. All observations with anomaly score not exceeding that threshold are taken as outliers. Note that this scoring rule has $\mathcal{O}(N \cdot M)$ complexity, where $M = \sum_{j=1}^{D} K_j$ and therefore scales linearly with sample size and the number of attributes (i.e., bins) which makes it a fast unsupervised anomaly detection algorithm.

## 3. Model Combination

Generally speaking any statistical model can be perceived as a (hopefully "useful") approximation to the (unknown) data generating process (see Box 1979). In the context of unsupervised learning the selection of the most useful base learner can be tricky due to the lack of labels (i.e., ground truth) which might help the statistician within a model selection task. This motivates using an ensemble of outlier detectors instead of a single detector since the former combines the strength of different base algorithms and thus can lead to an overall performance improvement. The basic idea of ensemble methods is that some models might do better on a particular subset of the feature space whereas others might do better on other regions of the feature space and therefore combining them could (although not necessarily in practice) lead to a boost in performance (cf. Zhou 2012).

For classification (see Dietterich 2000; Valentini and Masulli 2002) and clustering tasks (see Gosh and Acharya 2011) model ensembles are widely used and have a sound theoretical foundation. Methods like boosting, bagging, stacking etc. are nowadays standard ensemble strategies in practice as well as in research. Aggarwal (2012) provides a review of recent outlier ensemble methods and also highlights the relationship of ensemble analysis to the bias-variance trade-off. Because of the unsupervised nature of anomaly detection it is much harder though to reduce bias in outlier ensembles due to the absence of labels. Therefore the idea is to use an average of base detectors to reduce model-specific variance instead. Subsequently we will compare and discuss different approaches to outlier ensembles. From a practical point of view we have to answer the following questions: (i) which models out of a given candidate set of models, $\mathcal{M}$, should be selected for ensemble construction? (ii) How to normalize the different score distributions? (iii) And how to combine the different score distributions to an ensemble score distribution?

For the first question we will focus on two strategies, namely (1.) using a static (or full) ensemble (i.e., all models in $\mathcal{M}$) versus (2.) a dynamic ensemble construction, which selects a subset of candidates based on statistical criteria. For the latter we will use the "Greedy Ensemble construction" approach of Schubert et al. (2012). The two main requirements

for an ensemble to improve over its base models (and similar to ensembles in supervised learning tasks) are that the base learners themselves have high accuracy and are diverse at the same time, i.e., they make independent errors on new samples. The latter point should intuitively make sense because if the ensemble scores were on the other hand highly associated there would be no real improvement in predictive performance by combining them compared to just using a single base detector.

For score normalization there are mainly two strategies being used in the literature (cf. Aggarwal 2012): normalizing the base model scores so that their distributions are more comparable, e.g., by using the usual location/scale standardization ('z-transformation'). Alternatively if the original scores are of less importance (e.g., the actual score differences) and only the relative order of the samples is of interest then the underlying rank series can be used instead of the actual scores. For score combination it is common to use some sort of aggregation function, like the (weighted) average of the model component scores (or ranks) per sample, the median score or the minimum or maximum (depending on the model), see Zimek et al. (2014) for a general discussion of the challenges involved with outlier ensembles.

### 3.1. Static Ensemble Approach

In the model applications presented below we will use model averaging among other strategies for the prediction of anomalies in the data. The goal there is to apply a weighted average to the score distributions of different anomaly detectors: $p(S) = \sum_{j=1}^{J} \omega_j \cdot p(S|M_j), j = 1, \ldots, J$. From a statistical viewpoint the model weights should express model uncertainty related to the different candidates, i.e., the different errors committed by the candidate models. In a supervised learning setting we could use the underlying loss (or risk) function to measure predictive uncertainty of the different algorithms. In a Bayesian framework we could use the prior predictive distribution (or model marginal likelihood) to compute model weights based on model posterior probabilities (cf. Vosseler and Weber 2018). Unfortunately neither of this is applicable here so we are left with more "ad-hoc" approaches to model weight construction. The most obvious approach is to assign uniform ("uninformative") weights to the different ensemble components to express lack of prior knowledge. Another approach is to assign weights to the different models according to their score (or rank) similarity to other models, measured by some statistical association measures, e.g., Spearman's rank correlation. The problem with such an approach is however that a lack of ensemble diversity is being rewarded and one could clearly argue that this approach questions the model combination idea as a whole. In the most extreme situation of perfect association of the considered candidate models one could simply pick any arbitrary candidate to arrive at the same model performance as with the full ensemble, thus gaining no improvement at all but having higher computational training costs.

### 3.2. Dynamic Ensemble Approach

Rather than using all candidate models and combining them a more advanced approach is to employ a dynamic ensemble construction (cf. Campos et al. 2018; Zhao et al. 2019). Because of the absence of labels in the unsupervised setting it is common to construct a pseudo ground truth score vector, which can be converted to a binary vector to yield a pseudo target variable. The benefit of using a pseudo ground truth in an unsupervised setting is clearly that one can borrow ideas from binary classification to build model ensembles, e.g., Boosting (cf. Campos et al. 2018).

We will use the Greedy ensemble approach presented in Schubert et al. (2012); Zimek et al. (2014). In contrast to the mentioned works we will not use the weighted Pearson correlation on the ranks[7] to measure the similarity between two different outlier detectors, but pursue an information theoretic approach, by calculating the mutual information of their score distributions. The latter takes into account not only the first two moments of a

(score) distribution but also higher moments that capture shape. For two random variables $X$ and $Y$ the mutual information is defined as

$$I(x;y) = \int_{\mathcal{X}} \int_{\mathcal{Y}} p(x,y) \cdot \log\left(\frac{p(x,y)}{p(x) \cdot p(y)}\right) \cdot dx \, dy \tag{15a}$$

$$= E_{X,Y}\left(\log\left(\frac{p(x,y)}{p(x) \cdot p(y)}\right)\right) \tag{15b}$$

with $I(x;y) \geq 0$.

For two outlier detectors and their empirical score distributions expression (15) can be estimated by first estimating the joint and marginal densities, e.g., using a Gaussian mixture model,[8] and then taking a random sample from the fitted joint distribution to approximate the expectation in (15b) via basic Monte Carlo integration. Alternatively, one could estimate the above expression by using histogram estimators for the joined and marginal densities and then summing over the two-dimensional (binned) support. In our analysis below we will use the latter approach for computational reasons.

In Algorithm 2 the model selection approach of Schubert et al. (2012) with mutual information is sketched for an ease of reference.

Similarly to the original approach of Schubert et al. (2012) using a weighted measure of association we restrict the calculation of the mutual information to a balanced sample instead of using the full data set. By definition the "normal" ("majority or negative") observations will be dominating the sample and since we are more interested here in "anomalous" ("minority or positive") observations we a-priorily assume that $p\%$ (e.g., 5%) of the observations are positives.[9] To balance the computation of the mutual information we therefore only use the next $p\%$ of observations in a ranked series, i.e., the negatives that are close to being positives.[10]

---

**Algorithm 2:** Greedy Model Selection

---

$J :=$ Set of individual outlier detectors;
$K :=$ Union set of top-k outliers;
$v :=$ Target vector (=Pseudo ground truth);
/* with $v_i = 1$ if sample $i \in K$ and $v_i = 0$ otherwise */
$E := \emptyset$         /* Initialize ensemble */;
Sort $J$ by mutual information ($MI$) with $v$;
$E \leftarrow E \bigcup \text{getFirst}(J)$ ;
$S_E \leftarrow$ Combine scores in $E$ ;
Sort $J$ by mutual information with $S_E$;
**while** $J \neq \emptyset$ **do**
    $i \leftarrow \text{getFirst}(J)$;
    **if** $MI(E \bigcup i, v) > MI(E, v)$ **then**
        $E \leftarrow E \bigcup i$ ;
        $S_E \leftarrow$ Combine scores in $E$ ;
        Sort $J$ by mutual information with $S_E$;
        /* in decreasing order */
    **end**
**end**
**return** $E$;

---

### 3.3. Model Explanation

A crucial part of any deployed machine learning solution is model explanation, i.e., providing a human-friendly explanation to the user as to why an observation (here: a claim) was scored the way it was scored. Unfortunately most research around model explainability focuses on supervised learning problems (see Adadi and Berrada 2018;

Molnar 2022 for an overview) and hence cannot be used in the context of unsupervised anomaly detection. One recent exception is the work presented in Oliveira et al. (2021), however the latter work does not deal with ensembles but only with the case of a single anomaly detector and hence cannot be used here. For this reason we propose the following approach which is simple to implement and works well in practice. Recall that after having trained the outlier ensemble we have learned different (score) functions $\hat{f}_1(X), \ldots, \hat{f}_M(X)$ whose outputs are combined, e.g., using an arithmetic average ("pseudo target"), to yield a final ensemble score $\tilde{s}$ for observation $i$: $\tilde{s}(X_i) = \frac{1}{M} \sum_{m=1}^{M} \hat{f}_m(X_i)$, $i = 1, \ldots, N$. In order to explain the mapping $g : X \mapsto \tilde{s}(X)$ we train a global "surrogate" model, e.g., a linear regression. For example using a linear model for the explanation of individual score values $\tilde{s}_i$ we simply calculate the "effects", i.e., the weight per feature $j$ times the feature value of an instance: $effect_j^{(i)} = \beta_j X_{i,j}$, $j = 1, \ldots, J$. Individual effects of a feature, $effect_j^{(i)}$, could then be further compared to some summary statistics like the median or higher order statistics of the distribution of $effect_j$ (cf. Molnar 2022). In case a more complex surrogate model, like for example Light GBM, would be used to achieve a better fit we could simply apply any preferred model explanation method like SHAP (see Lundberg and Lee 2017) subsequently to get individual score explanations. To our knowledge, this approach although simple and straightforward to implement has not yet been proposed in the outlier detection literature. Independently from the above Zhao et al. (2021) recently proposed an approach to accelerate the scoring on new data points with a large number of unsupervised, heterogeneous outlier detectors. The authors use a similar idea like us, i.e., approximating unsupervised outlier detectors by supervised regression models using the predicted values as pseudo ground truth. However their focus is to achieve faster offline prediction in outlier ensembles and not to enable local ensemble explainability and therefore they train a surrogate model for each single outlier detector rather than for the whole ensemble as we do.

Another approach that we used in practice (although not as generally applicable as the above) was to utilize the above presented BHAD also as a model explainer ("surrogate model"). Recall from Section 2 that due to the linearity of the model we can simply check which terms in expression (14) contribute the most to the individual anomaly score value of observation $i$. Although this seems like a brute-force approximation to the actual ensemble model we could show at least empirically that for many used data sets the BHAD would be highly correlated with the other candidate models in the ensemble and hence using it as an approximation seems not too restrictive. A user could then be presented the five most influential features per observation $i$ alongside with the associated values $\hat{F}_j(x_{i,j})$ where $\hat{F}_j$ is the empirical cumulative distribution function of feature $j$.

We will next study the predictive performances of the discussed models in a Monte Carlo experiment and also using two popular benchmark datasets for outlier detection.

## 4. Simulation Experimental Design

As a data generating process (DGP) we use a two-component multivariate Student-$t$ mixture distribution. Let $z_i \in \{0, 1\}$, $i = 1 \ldots N$, be the latent assignment of observation $i$, where 1 means fraudulent and 0 otherwise. Further let $\pi \equiv \text{Prob}(z_i = 1)$ be the probability of being an outlier. The generative process is given by:

$$z_i \sim \text{Bern}(\pi) \quad \text{for } i = 1, \ldots, N \tag{16a}$$

$$\Sigma^{(j)} \sim W_p(\mathbf{V_0}, \nu^{(j)}) \tag{16b}$$

$$X_i | (z_i = j, \Sigma^{(j)}) \sim N_p(\mu_0^{(j)}, \Sigma^{(j)}) \quad \text{for } j = 1, 2 \tag{16c}$$

Here $\text{Bern}(.)$ denotes the Bernoulli distribution, $W_p(.)$ denotes the $p$-dimensional Wishart distribution with $\nu^{(j)} > p - 1$ degrees of freedom and known $p \times p$ symmetric

positive semi-definite scale matrix $\mathbf{V_0} > 0$. Finally $N_p(.)$ denotes the $p$-dimensional Normal distribution with component-specific first and second moments.

As base learners for the outlier ensemble we use the following models: Variational Autoencoder (VAE) with Gaussian prior (see Kingma and Welling 2014), VAE with stick-breaking prior (i.e., a Dirichlet process) (SB-VAE) (see Nalisnick and Smyth 2017), the Bayesian histogram anomaly detector (BHAD) of Section 2, Isolation forest (Liu et al. 2012), One-class SVM (OCSVM), average k-Nearest Neighbors (kNN)-based outlier detector, Angle-based Outlier Detector (ABOD) (Kriegel et al. 2008) and the Local Outlier Factor (LOF) (Breunig et al. 2000).

Note: although VAEs were originally not developed for the task of anomaly detection, but rather for compression or dimensionalty reduction, they can easily be used for this purpose, cf. An and Cho (2015); Chen et al. (2018); Gomes et al. (2021). For this we re-interpret the reconstruction error of the variational lower bound on the marginal likelihood of a data point $i$ (see Equation (3) in Kingma and Welling 2014) as an anomaly score for that observation. This means that sample points that have a high reconstruction error associated are declared to be anomalous compared to other points.

After having prepared the training data[11] we independently train each of the candidate models. For ensemble construction we compare the greedy model selection with mutual information (see Algorithm 2) with the greedy model selection with weighted Pearson correlation using boosting (see Campos et al. 2018) and a full ensemble, i.e., using all candidate models. In terms of model combination we use a simple arithmetic average of the model score ranks as well as the minimum model rank per observation. In our Monte Carlo experiments we use $\pi = 0.01$ for $p = 30$ with $\mu_0^{(1)} = -1$, $\mu_0^{(2)} = 0.5$, $\mathbf{V_0} = \mathbf{1}_p \cdot 1.5$, $\nu^{(j)} = 3p, \forall j$ and take $N = 30{,}000$ random draws from the DGP in (16). For each sample $\{x_i\}_{i=1}^{N}$ we train an outlier ensemble and repeat this $M = 100$ times. Figure 3 shows a sample of the DGP projected along the first three singular vectors:

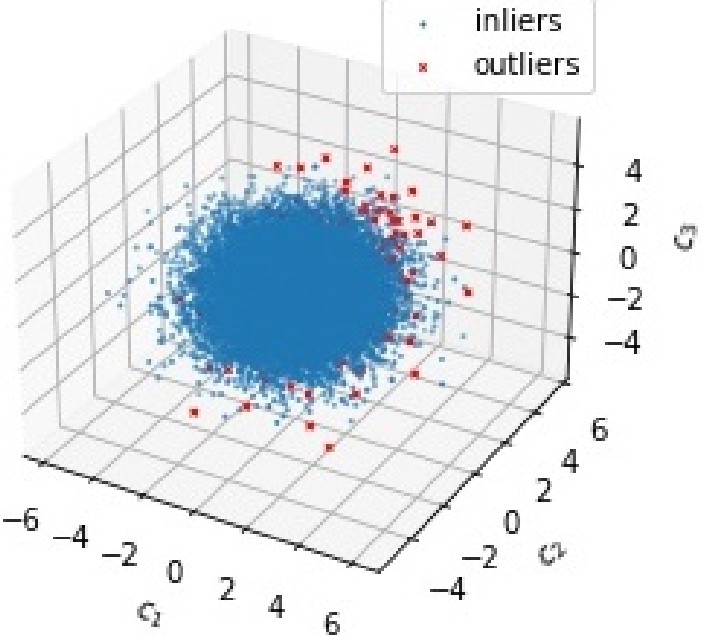

**Figure 3.** Draw from the mixture DGP (compressed).

In our model trainings we use the following (hyper) parameter settings:

- Gaussian VAE: we choose 80 training epochs, learning rate of 0.001, latent space of dimension 10 and a single fully connected hidden layer with 200 nodes in the encoder and decoder network (see Kingma and Welling 2014 for details)
- SB-VAE: we choose $\alpha = \beta = 2$ as shape hyperparameters of the Beta prior p.d.f. used in the stickbreaking algorithm, a learning rate of 0.001 and the network architecture as for the VAE (see Nalisnick and Smyth 2017 for details).[12]
- Bayesian histogram-based anomaly detector (BHAD) using $K_{Max} = 60$
- Isolation Forest with $M = 200$ number of trees
- One-class SVM with radial basis function kernel
- Average k-Nearest Neighbors over $k = 2, 4, 6, 10$
- Angle-based outlier detector (ABOD) using $k = 10$
- Local Outlier Factor (LOF) using $k = 10$
- Greedy algorithm with weight boosting using a drop rate $d = 0.3$ (see Campos et al. 2018)

The results of the Monte Carlo experiment are shown in Table 1. There the Monte Carlo estimates of F1 score, Precision, Recall and the area under the ROC curve (AUC) are reported for each of the single models as well as for different ensemble approaches. Note that since we know the DGP and hence the true "state of nature" we can evaluate the different models as we would do in a binary classification setting. To correct for the class imbalance we calculate F1, precision and recall based on a re-weighted sample, with sample weight of observation *i* being inversely proportional to the relative frequency of its class.

**Table 1.** Results—Monte Carlo experiment.

| Model | F1 Score | Precision | Recall | AUC |
|---|---|---|---|---|
| SB-VAE | 0.9791 | 0.9996 | 0.9598 | 0.9998 |
| VAE | 0.9566 | 0.9991 | 0.9180 | 0.9977 |
| BHAD | 0.9715 | 0.9995 | 0.9455 | 0.9996 |
| IForest | 0.9437 | 0.9989 | 0.8947 | 0.9986 |
| OCSVM | 0.9390 | 0.9987 | 0.8865 | 0.9974 |
| kNN | 0.9828 | 0.9997 | 0.9671 | 0.9999 |
| ABOD | 0.9533 | 0.9924 | 0.9176 | 0.9966 |
| LOF | 0.9723 | 0.9999 | 0.9477 | 0.9999 |
| Full ensemble [1] | 0.9735 | 0.9995 | 0.9493 | 0.9997 |
| Minimum rank [2] | 0.9640 | 0.9993 | 0.9317 | 0.9997 |
| Greedy MInfo [3] | 0.9817 | 0.9997 | 0.9651 | 0.9998 |
| Greedy Pearson [4] | 0.9828 | 0.9997 | 0.9671 | 0.9999 |

[1] Full ensemble: Arithmetic model average with uniform weights; [2] Minimum rank: Minimum model rank per observation; [3] Greedy MInfo: Model selection using mutual information; [4] Greedy Pearson: Model selection using weighted Pearson correlation with Boosting.

From Table 1 we can see that all base learners are able to detect the true outliers in the data very well, which was somehow to be expected given the simplicity of the synthetic data set. Furthermore the results suggest that the proposed Bayesian anomaly detector (BHAD) achieves very competitive results compared to other more established methods. Although the more computationally expensive SB-VAE and k-NN perform slightly better than BHAD in terms of F1 score and AUC this difference is not statistically significant according to an approximate paired *z*-test for the difference in proportions as well as a McNemar-Test (see Dietterich 1998) with significance level 5%. Comparing the performances of the two versions of the Greedy model selection in terms of F1 score and AUC with the other ensembles and single candidates it can observed that the Greedy algorithm achieves a better predictive quality, except for the k-NN model, which is the best performing approach here (although the difference to the Greedy ensembles is again not significant).

To analyze the diversity of the different ensemble components we can calculate the similarity matrix using, for example, Spearman's rank correlation $\rho$ (see Figure 4). From there we can observe that the candidate models although quite different from a statistical viewpoint are relatively strongly connected. In particular the overall high correlation of the BHAD with the other base detectors should be noticed (in average: 0.91). One way to further improve diversity among the different candidate models is to train the models on various hyperparameter settings.

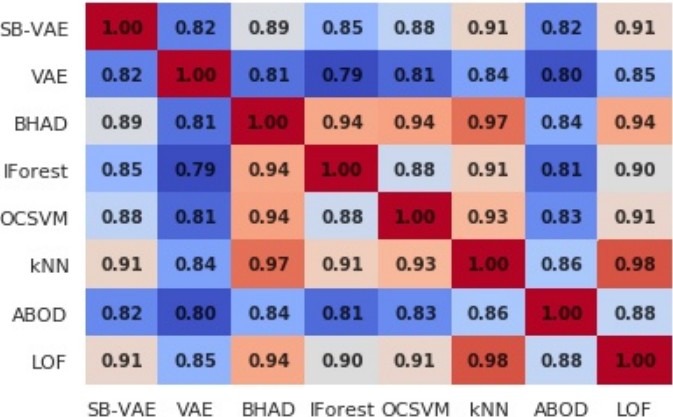

**Figure 4.** Spearman's $\rho$ of score distributions.

## 5. Application: Benchmarks Datasets

Next we will apply the same set of candidate models and ensemble approaches to two popular benchmark datasets in the literature on outlier detection (cf. Schubert et al. 2012): (1.) the pendigits (Pen-Based Recognition of Handwritten Digits) dataset and (2.) the Wisconsin-Breast Cancer (Diagnostics) dataset (WBC). Both datasets are originally taken from the UCI machine learning repository. The pendigits dataset is originally a multiclass classification dataset having 16 integer attributes and classes 0, ..., 9. In this dataset, all classes have equal frequencies, so the number of objects in one class (corresponding to the digit "0") is reduced by a factor of 10%. On the other hand the Wisconsin-Breast Cancer dataset is originally a classification dataset, which records the measurements for breast cancer cases. There are two classes, benign and malignant. The malignant class of this dataset is downsampled to 21 points, which are considered as outliers, while points in the benign class are considered inliers.[13]

In our benchmark analysis we draw a stratified sample of 80% of the data points (without replacement). As stratum we use the binary target variable (0: no anomaly, 1: anomaly). We train the models on this random sample and calculate the same performance metrics as used in Section 4 above. This was repeated $M = 100$ times until finally we calculate average performance metrics over all Monte Carlo runs.

The results are depicted in Tables 2 and 3, respectively. For the pendigits data (Table 2) it can be seen from the AUC column that although the two versions of the greedy algorithm give very good results, a single Isolation forest is actually slightly better. On the other hand BHAD is only slightly worse than the Isolation forest, however a McNemar test indicates that the difference in performance is not statistically significant.

For the WBC dataset a single BHAD is the best performing model alongside with an ensemble model with Greedy model selection.

**Table 2.** Results: Pendigits dataset.

| Model | F1 Score | Precision | Recall | AUC |
|---|---|---|---|---|
| SB-VAE | 0.6670 | 0.5030 | 0.9901 | 0.6691 |
| VAE | 0.6669 | 0.5027 | 0.9904 | 0.6452 |
| BHAD | 0.7767 | 0.6523 | 0.9600 | 0.9410 |
| IForest | 0.7900 | 0.6714 | 0.9609 | 0.9508 |
| OCSVM | 0.7004 | 0.5532 | 0.9542 | 0.8304 |
| kNN | 0.6838 | 0.5333 | 0.9525 | 0.7290 |
| ABOD | 0.6727 | 0.5237 | 0.9401 | 0.6714 |
| LOF | 0.6731 | 0.5143 | 0.9739 | 0.5125 |
| Full ensemble | 0.7022 | 0.5556 | 0.9543 | 0.8515 |
| Minimum rank | 0.7245 | 0.5832 | 0.9563 | 0.8635 |
| Greedy MInfo | 0.7759 | 0.6512 | 0.9601 | 0.9412 |
| Greedy Pearson | 0.7766 | 0.6520 | 0.9601 | 0.9409 |

**Table 3.** Results: Wisconsin-Breast Cancer (Diagnostics) dataset.

| Model | F1 Score | Precision | Recall | AUC |
|---|---|---|---|---|
| SB-VAE | 0.6941 | 0.5329 | 0.9954 | 0.8924 |
| VAE | 0.6703 | 0.5071 | 0.9886 | 0.7556 |
| BHAD | 0.8152 | 0.6988 | 0.9793 | 0.9485 |
| IForest | 0.7967 | 0.6734 | 0.9763 | 0.9390 |
| OCSVM | 0.7225 | 0.5778 | 0.9642 | 0.6693 |
| kNN | 0.7797 | 0.6507 | 0.9734 | 0.9252 |
| ABOD | 0.7295 | 0.5912 | 0.9539 | 0.9027 |
| LOF | 0.6865 | 0.5278 | 0.9823 | 0.7903 |
| Full ensemble | 0.7183 | 0.5720 | 0.9660 | 0.9113 |
| Minimum rank | 0.7608 | 0.6242 | 0.9749 | 0.9091 |
| Greedy MInfo | 0.8089 | 0.6882 | 0.9821 | 0.9482 |
| Greedy Pearson | 0.8105 | 0.6902 | 0.9824 | 0.9485 |

Next we will present some results of a fraud detection project from the domain of corporate insurance.

## 6. Application: Detection of Fraudulent Insurance Claims

One popular application of outlier/anomaly detection in the insurance industry is claims fraud detection. A possible definition from the context of corporate insurance is that claims fraud is considered as *"the intentional deception and/or material misrepresentation to another party about an insurance matter in order to receive money or other benefits which are not rightfully theirs. Fraud occurs when an insured or a third-party deliberately misrepresents a loss or part of a loss that, if true, would be covered by an insurance policy, but which in fact did not occur."*[14]

In the following we will focus on an application in the context of the corporate insurance domain. In practice upon establishing an Anti-Fraud business process it is very common that there has not been any fraudulent claims detected yet, hence the need for unsupervised learning methods. The lack of valid fraud labels puts an increased emphasis on the feature engineering process. Therefore we will first dive a bit deeper in the construction of relevant fraud features to capture potential fraudulent claims information.

### 6.1. Feature Engineering

Feature extraction in general is highly domain specific, for example, in a life insurance scenario, a low lag between the policy start date and the death of the subject is sometimes correlated to homicide, whereas in other insurance domains like corporate insurance it might be indicative for fraudulent behavior.

In our construction of relevant features we view a claims event according to different informational layers or dimensions.

In Table A1 of the Appendix A the list of model features used in our application is shown. As can be seen a larger group of features was derived from the claim description, i.e., from free text in which the claims handler briefly describes what has happened. All features named "Claim description contains xxx and similar" were constructed as follows: after having preprocessed each document[15] (i.e., removal of stopwords, tokenization etc.) we use pretrained GloVe word embeddings (see Pennington et al. 2014)[16] We also tried training custom word embeddings on the claim descriptions using Word2vec, Doc2vec, tf-idf and FastText, however since the used claim descriptions are often rather short (sometimes only two or three words) choosing a transfer learning approach, utilizing a model trained on a large text corpus gave better results. We then aggregated the 300 dimensional GloVe token embeddings by simple arithmetic averages to a document embedding for each claim description. Since we want to construct a numeric feature that captures the semantic similarity of a claim description with a given list of keywords "xxx" (e.g., "Claim description contains *water, damage, broken, repair* and similar"), we also assign the word embeddings to each token in the key word list and then use a similarity measure (e.g., cosine similarity) to measure how "semantically" close the two vector representations are to each other.

Another interesting group of features can be motivated as follows: often it is beneficial in the feature engineering process to control for (un)intended misspellings in unstructured data like names, email addresses, telephone numbers, locations etc. From a data quality perspective but also from a data analysis perspective we do not want to treat two terms like "Apple" and "Appl" independently as they should actually be treated as one. For this reason we developed a simple but effective approach for fuzzy names matching and applied it to the insured names and claimant names to subsequently work with, e.g., a (fuzzy) grouped insured name. For the fuzzy names matching we first take a corpus of names (but it could be applied essentially to any character sequence without minor modifications) and first convert each name to character n-grams ($n = 3$ gives good results in most applications) after having preprocessed the data. Then based on the character n-grams we fit a tf-idf model and represent each insured name by its tf-idf embedding. This yields a $n \times k$ matrix $D$, where $n$ is the number of insured names and $k$ the dimension of the tf-idf embedding. Using this matrix we calculate next an $n \times n$ distance matrix, which we finally use in a DBSCAN clustering algorithm (see Ester et al. 1996). Many of the resulting groups will be noise points (in the DBSCAN terminology) here, and hence form a singleton cluster. This makes intuitively sense since misspellings should not occur so often. However the interesting cases are the non-singleton clusters and these are then used instead of the original names in further course of the analysis. In the list of used features in Table A1 the term "per group" therefore refers to using the clusters from a fuzzy names matching step instead of the original names.

Finally another group of features we would like to highlight here are based on the concept of rolling time windows in order to capture anomalies (i.e., breaks) over time. For example, the number of claims associated with $A$ in the last $Y$ days before date $X$. We created several features of this kind for $A$ = Insured, Claimant, Payee. For example: number of claims for a particular policy within 90 days ("Rolling number of claims per policy"). These features are also constructed based on the above described fuzzy names matching groups.

Next we will present the results of a fraud detection project from the domain of corporate insurance.

### 6.2. Empirical Results

In our empirical application we follow the same approach as outlined in the simulation design above, i.e., we first train all models separately and then we use the greedy model selection approach with mutual information. As a final "score" for observation $i = 1, \ldots, N$ we use the arithmetic average $\bar{r}_i = \frac{1}{J} \sum_{j=1}^{J} r_{i,j}$ of the rank $r_{i,j}$ per candidate model $j$.[17] Working with rank statistics instead of the original scores has the advantage that we

directly work with a normalized quantity and also that ranks can be easily understood by less technical users.

For didactic reasons when presenting the model output to non-technical users we choose to further map the averaged outlier ranks to a uniform discrete scale $1, 2, \ldots, 5$, where a value of 5 indicates high fraud potential. For the mapping we induce a skewed distribution shown in Figure 5 with relative frequency for a score of 5 equal to 1%, i.e., the used contamination rate in all models.

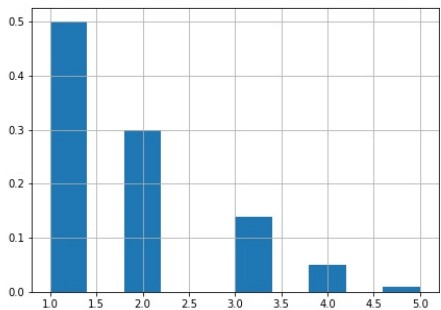

**Figure 5.** Distribution of final fraud score.

As mentioned above at the time of writing we only had a very limited number of observed fraud cases and no validated non-fraud cases available. This is the reason why we focus on our model evaluation on the recall rather than on criteria that require positively-negatively labelled data (like the F1 score). Also what should be noted is that in the presented business application the actual use case is not directly fraud detection, but rather more detecting "fraud referrals". This means we want to rank a claims portfolio according to our model and then propose the user a small subset of cases, namely the cases that have a score of 4 (suspicious) or 5 (highly suspicious). Since we have limited ground truth we focus on the evaluation of the recall (or hit rate) ($= \frac{TP}{TP+FN}$), with $TP$ the number of true positives and $FN$ the number of false negatives. In our case we declare a claim to be a "positive" case if it's among the referred cases of a dedicated fraud investigation team. Table 4 depicts the model performances based on the collected labels (i.e., referrals) at the time of writing. In the last column we also show the corresponding Bayesian highest posterior density interval of the (unknown population) recall. The latter can be calculated by assuming that the number of true positives follows a Binomial distribution, $\mathrm{Bin}(p)$, and using a conjugate Beta prior for the unknown probability $p$ that claim $i$ is correctly predicted given it is fraud (or at least a referral). Then using the resulting Beta posterior we calculate the 95% highest posterior density interval to express uncertainty associated with the reported numbers, e.g., due to small sample sizes.

**Table 4.** Results—Insurance claims fraud detection.

| Country | Recall/Sensitivity | 95% HPD [1] interval |
|---|---|---|
| UK | 0.26 | [0.19, 0.36] |
| US | 0.14 | [0.13, 0.16] |
| Spain | 0.10 | [0.02, 0.41] |
| Germany | 0.55 | [0.37, 0.72] |
| South-Africa | 0.07 | [0.02, 0.32] |

[1] HPD: Highest posterior density.

From Table 4 we observe that the recall varies between 7% and 55%, where the quite different sample sizes per country are reflected in the width of the HPD intervals of $p$. The variation in measured performances can also be attributed to the underlying data collection process. The different countries have quite different processes when it comes to fraud prevention and hence the way referrals get gathered are not driven by a homogeneous

business process, which makes it more challenging to compare the results directly. Another challenge in many other business applications is that some information accessible to a human fraud expert are in paper files or email correspondences that in the end may drive the decision towards or against being referred for further investigation or not. The current model still produces quite some false positives (based on business feedback), but very often the reason for not agreeing with the model's prediction is outside of the model scope (i.e., not a part of the training data), so this remains a challenge in fraud applications.

## 7. Conclusions

The task of anomaly detection has many important industrial applications with fraud detection in insurance being one of the most popular ones. For the latter we presented an approach based on outlier ensembles using a variety of algorithms which were combined to produce a final more powerful model. We also introduced BHAD, a Bayesian histogram-based anomaly detector, which achieved very competitive predictive results compared to other more complex models based on a Monte Carlo experiment as well as on two popular benchmark datasets. BHAD has linear complexity with respect to the sample size and the number of attributes and allows a straightforward explanation of individual anomaly scores. It can be used for both continuous and discrete features, where in the latter case obviously no binning is required. One critical issue with the BHAD is its feature independence assumption, which might not be appropriate for some data sets. Another debatable point is that BHAD assumes the knot positions of the histogram as known rather than treating them as unknown parameters. For example, if interest was in inferring the knot locations within a Bayesian model selection framework one could use a similar approach as in Vosseler (2016) using MCMC. An alternative route here that has actually a long tradition in non-parametric Bayesian statistics (cf. Müller et al. 2015) is to treat the unknown probability distribution *G* of the data as a random probability measure to which a prior probability model can be assigned. A commonly used prior for *G* here is a Dirichlet process. An anomaly detection score could then be constructed in a similar way as for the BHAD (at least in principle) based on the estimated density values.

We also compared a variation of the greedy algorithm of Schubert et al. (2012) using mutual information of two score distributions with the original approach using Pearson correlation. Overall we did not find evidence that using the mutual information criterion results in a significant improvement gain. For model explanation in unsupervised outlier ensembles we also proposed a model-agnostic approach, which approximates the ensemble score by a supervised surrogate model. In principal, this approach can be utilized for the explanation of real-valued anomaly scores (regression) as well as of binary anomaly predictions (classification) in the same way.

Lastly we applied the presented methods to the problem of detecting fraudulent insurance claims in the corporate domain. For some countries the model's sensitivity looks promising while for others yet not so much. Apart from the already discussed data quality issue regarding reliable labels there are some routes that are promising for improving the model. One obvious choice would be to make the "external" information accessible to the model via additional features. Another potential future improvement is to represent the claims data as a multidimensional undirected graph and then using standard network metrics to derive additional model features. As more labels are being collected it makes sense to also utilize this supervision to increase the model's accuracy. For this reason one alternative to the fully unsupervised approach presented here could be to use a positive-unlabelled (PU) learning approach (see Elkan and Noto 2008) or a semi-supervised approach once valid negative instances become available.

**Funding:** This research received no external funding.

**Institutional Review Board Statement:** Not applicable.

**Informed Consent Statement:** Not applicable.

**Data Availability Statement:** Details regarding data supporting the reported results can be found at http://odds.cs.stonybrook.edu/.

**Acknowledgments:** The author would like to thank all the persons at AGCS who have contributed to the project over the years: Daniel Didt, Ahmad Bin Qasim, Uzair Akbar, Jiaqi Lu, Roman Marchuk, Roxana Mirshahvalad, Gareth Davies and Yifeng Lu.

**Conflicts of Interest:** The author declares no conflict of interest. Views and opinions expressed are his own and not those of Allianz Global Corporate & Specialty SE.

## Appendix A

**Table A1.** Used fraud model features.

| Short Description | Type |
| --- | --- |
| Diff. in days—notification date and date of loss | numeric |
| Years of client relationship | numeric |
| Number of claims per policy | numeric |
| Rolling number of claims per policy | numeric |
| Number of claims per claimant | numeric |
| Rolling number of claims (insureds) per group of claimants | numeric |
| Number of claims per insured | numeric |
| Loss ratio per insured (in percentage) | numeric |
| Claim description contains keywords for accidents and natural causes | numeric |
| Claim description contains collision, impact, crash, sink, grounding and similar | numeric |
| Claim description contains fire, burn and similar | numeric |
| Claim description contains luggage, baggage, cash, money, item, belonging and similar | numeric |
| Claim description contains passenger, bodily injury, poisoning and similar | numeric |
| Claim description contains plane, aircraft, helicopter and similar | numeric |
| Claim description contains storm, wind, weather and similar | numeric |
| Claim description contains stolen, theft,disappear and similar | numeric |
| Claim description contains vessel, ship and similar | numeric |
| Claim description contains water, damage, broken, repair and similar | numeric |
| Claim description contains yacht and similar | numeric |
| Claim description contains similar fraud keywords | numeric |
| Diff. in days—date of policy expiration and date of notification | numeric |
| Various interaction variables of claim amount and combinations of above features | numeric |
| Date of loss is public holiday | categorical |
| Date of notification is public holiday | categorical |

## Notes

1    Note expression (1) constitutes a proper density function.
2    Note by simply replacing the probabilities $\pi_k$ by their relative frequency counterparts $n_k/n$ yields the basic histogram density estimator $\hat{f}_H(y_i)$, cf. Scott (2015).
3    We will omit conditioning on $\xi$ in the following to keep notation simple.
4    For comparison: frequentist estimates using Sturges rule are $\hat{K} = 30$ and $\hat{K} = 96$ using Freedman-Diaconis rule.
5    With scalar $y_i$ replaced by scalar $x_{i,j}$ in the following.
6    Obviously the AVF algorithm could easily extended to continuous data, by simply using a discretization, like binning.
7    This is equivalent to the non-parametric Spearman's $\rho$ coefficient.
8    The only restriction on the selected density model is that we should know how to generate draws from it.
9    This is sometimes called the contamination rate in the literature.
10   This puts both groups of observations on the same footing since otherwise the mutual information would be driven by the vast majority of non-outlier cases.
11   Here all features are normalized to having zero mean and variance of one.
12   We used the publicly available Theano code of the authors and integrated it into a generic scikit-learn Python class API.
13   Both datasets are available online from http://odds.cs.stonybrook.edu/ (accessed on 1 May 2022).

14    This is the AGCS SE standard on Anti-Fraud issued by AGCS compliance.

15    Subsequently each claim description corresponds to a "document".

16    These are easily accessible via Python libraries like *spaCy*.

17    Setting $\min_i(\bar{r}_i) = 1$ and $\max_i(\bar{r}_i) = N$ in case needed.

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
