# Peer review of "Unsupervised Insurance Fraud Prediction Based on Anomaly Detector Ensembles"

_risks, doi:10.3390/risks10070132_

Round 1

Reviewer 1 Report

This article considers the problem of using an outlier ensemble to predict fraud in the insurance business. The problem is interesting. The unsupervised learning method is also interesting. the paper is on the topic of the special issue.

The mathematics in this paper is standard, and the method in the paper is also well known.  The paper is well-written in general. I cannot say that the English are poor, but I think it can be improved. The literature review is not comprehensive. For example, the paper below considered a similar problem and used some similar methods (unsupervised learning, also the related references in the paper):

C. Gomes, Z. Jin, and H. Yang, ``Insurance Fraud Detection with Unsupervised Deep Learning", Journal of Risk and Insurance, Vol. 88, 591 - 624, 2021.

Reviewer 2 Report

This paper presents a Bayesian histogram anomaly detector (BHAD), where the number of bins is treated as an additional unknown model parameter with an assigned prior distribution. This is an interesting work. The structure is well organised. 

Minor comments: you need to bring out your novelty of your work. 

Reviewer 3 Report

The abstract is way too long and not directly to the importance of the raised problem. The author is suggested to directly and clearly state what the problem is, why it is important, and briefly state what is the novelty of the proposed work. I read through it a couple of times and was not convinced what the novelty is. The introduction section though made it clear.

The author is suggested to state why the geometric prior probability was chosen (Eq. 4). 

The author is recommended to use "U" as the notation of uniform instead of \mathbf{1} in (5)

It would be good for the reader to see how (8a) turned into (8b). Is the result of (8b) proportional to (8a)?

The quality of Fig. 4 is a bit poor. May be using a bold font for the numbers?

Page 11, line 306: Recommend using \text{Bern}(.) rather than \text{Bern}(\pi)

I think this is a good manuscript for this journal. however, the author is suggested to address the raised questions and resubmit the revised version.

Round 2

Reviewer 3 Report

The author made acceptable changes to the manuscript based on the raised comments in the first round of review. I have no more comments about the technicality of the paper. However, the paper needs to go through a careful proofreading as there are numerous grammatical issues with the manuscript.  highly recommend the author to fix those before the final decision.